# The Impact of Exercise Cognition on Exercise Behaviors: The Mediating Role of the Satisfaction of Basic Psychological Needs in Exercise for Adolescents

**DOI:** 10.3390/bs14070574

**Published:** 2024-07-06

**Authors:** Jianhua Yan, Haiwei Ren, Congshuai Wang, Ke Zhou, Xiaofen D. Hamilton

**Affiliations:** 1School of Physical Education, Henan University, Kaifeng 475001, China; 10180106@vip.henu.edu.cn (J.Y.); 1443068347@henu.edu.cn (H.R.); 2School of Track and Field Sports, Beijing Sport University, Beijing 100084, China; 2022240986@bsu.edu.cn; 3Department of Curriculum and Instruction, The University of Texas at Austin, Austin, TX 78712, USA; xk93@austin.utexas.edu

**Keywords:** cognitive behavior, youth sports perceptions, middle school students, psychological needs fulfillment

## Abstract

In recent years, the exercise behavior of Chinese adolescents has been on the decline, which is extremely detrimental to their physical and mental health development. However, few studies have explored the mechanisms by which exercise cognition influences Chinese adolescents’ exercise behavior. The present study aimed to investigate the relationship between exercise cognition and exercise behavior among Chinese adolescents and the mediating role of satisfying basic psychological needs for exercise. The study consisted of 996 adolescents (44.6% males, 55.4% females) between the ages of 12 and 15 (M = 13.34, SD = 1.059). Participants’ exercise behaviors and the satisfaction of basic psychological needs for exercise data were collected via surveys. Structure equation modeling (SEM) was performed to examine the direct and indirect effects. The results were as follows: (1) adolescents’ perceptions of exercise were significantly associated with exercise behavior and (2) the mediation model suggests that the satisfaction of basic psychological needs for exercise is an important mechanism by which exercise cognition influences the occurrence of exercise behavior. Therefore, it is crucial to help adolescents form good exercise cognition. Schools, families, and society should take responsibility for adolescents’ formation of good exercise cognition and satisfy adolescents’ basic psychological needs for exercise so as to enhance adolescents’ exercise behaviors and thereby develop good exercise habits.

## 1. Introduction

The results of the 2019 Chinese Students’ Physical Fitness and Health Survey indicated that obesity and myopia rates remain high, and adolescents have not yet developed good physical activity habits despite improvements in their overall physical fitness and health status [1]. Therefore, it is particularly important to explore the factors that influence adolescents’ exercise behaviors. According to cognitive behavior theory, individuals are more likely to participate in an activity as their understanding of the behavior or its correlated outcomes intensifies [2]. For instance, greater knowledge and understanding of fundamental exercise rules and skills have been found to positively influence the development of exercise consciousness, thereby enhancing exercise behaviors [3]. Positive evaluations and recognition of the value and benefits of exercise are pivotal in fostering exercise behaviors [4]. Nevertheless, existing studies have primarily focused on external factors such as parental role modeling, encouragement, and support in adolescents’ exercise behaviors while neglecting to explore the potential impact of individual exercise cognition on exercise behaviors [5]. Furthermore, limited attention has been given to internal psychological factors (e.g., emotion, cognition, motivation, or attitude) and their influence on exercise behavior across all education levels. Thus, to contribute to the understanding of exercise behavior among adolescents, our study aimed to investigate the influence of exercise cognition and the satisfaction of psychological needs on adolescents’ exercise behaviors. It is hoped that this study will enrich our understanding of adolescents’ exercise behaviors and provide baseline data for promoting the holistic development of their physical and mental well-being.

### 1.1. Exercise Cognition and Exercise Behaviors

Cognition, a term rooted in psychology, refers to the process of acquiring knowledge or understanding [6]. Exercise cognition specifically pertains to our understanding of physical movement and includes beliefs about the benefits and barriers associated with physical exercise [7]. Generally, exercise cognition serves as the foundation for exercise consciousness and one’s capacity to act, where increased knowledge and understanding contribute to the development of higher-level exercise consciousness, thereby facilitating exercise behaviors [8]. Previous research has indicated a positive association between individual exercise cognition and exercise behaviors [9,10]. Those with more exercise-related cognitive errors tend to exercise less [11]. Additionally, a prior study has found that cognitive processes in exercise mediate the relationship between visual perception and proficient action [12]. This explains why elite athletes can circumvent information processing limitations when performing complex exercise skills. In summary, exercise cognition is widely considered a significant factor influencing exercise behaviors [13]. However, previous studies have paid less attention to adolescents’ exercise cognition, particularly regarding its impact on their exercise behaviors. Therefore, we posit the following:

**Hypothesis 1**:
*The exercise cognition of adolescents could positively influence their exercise behavior.*


### 1.2. Basic Psychological Need Satisfaction in Exercise and Exercise Behaviors

In addition to exercise cognition, basic psychological needs in exercise are also important factors affecting exercise behaviors. These needs manifest as higher self-determined motivation when met, leading to increased levels of exercise behavior [14]. The basic psychological needs in exercise are associated with self-determination theory (SDT), which encompasses various sub-theories [15]. Overall, SDT comprises three fundamental psychological elements: autonomy, competence, and relatedness. Vallerand and colleagues applied SDT to the study of exercise behaviors, highlighting that satisfying these three basic psychological needs is essential for internalizing exercise motivation and drive, collectively termed the basic psychological needs in exercise [14,16]. Specifically, the authors noted that the fulfillment of autonomy and competence contributes to the formation of autonomous motivation, which significantly impacts individual behavioral initiative and enthusiasm. Additionally, interpersonal factors, such as peer support and exercise communication, fulfill the need for relatedness, the third element of SDT, and play a vital role in shaping individual exercise behaviors. Greater satisfaction in relationships has been found to strongly influence one’s engagement in exercise [17]. Thus, when an individual’s basic psychological needs in exercise are met, it positively contributes to the emergence and development of his/her exercise behaviors [18]. Rubio et al. further pointed out that, after more than two decades of development, SDT can be applied to research in the field of exercise, encompassing various aspects such as behavior, motivation, emotion, and intention [19]. In other words, the higher the satisfaction of individuals’ basic psychological needs (i.e., autonomy, competence, and relatedness) in exercise, the more likely positive exercise behaviors will be generated.

Further research on adults has shown that motivation can significantly predict exercise behavior based on age, gender, and the satisfaction of basic psychological needs in exercise [20]. In particular, gender was found to moderate the relationships between motivational regulations and exercise, as well as between motivational regulations and psychological need satisfaction [21,22]. However, previous studies have primarily examined the satisfaction of basic psychological needs for exercise as a factor influencing adolescents’ exercise behaviors. The role of gender in adolescents’ satisfaction of basic psychological needs in exercise remains unclear. Meanwhile, scholars have suggested that future research should investigate gender differences in adolescents’ physical activity autonomy motivation [23]. Therefore, we posit the following two hypotheses:

**Hypothesis 2**:
*There were gender differences in exercise cognition, satisfaction of basic psychological needs in exercise, and exercise behavior among adolescents.*


**Hypothesis 3**:
*The satisfaction of basic psychological needs in exercise among adolescents could positively influence their exercise behavior.*


### 1.3. Basic Psychological Need Satisfaction in Exercise and Exercise Cognition

Further research has shown that the satisfaction of basic psychological needs in exercise is primarily a process in which an individual gradually achieves self-actualization based on their perceptions, mainly reflected in the prediction of behavior [24]. Researchers have also reported that exercise cognition is significantly associated with autonomy and competence during exercise. The satisfaction of basic psychological needs, such as autonomy and competence, further affects exercise motivation [25]. Additionally, psychological needs, as mediating variables, play a crucial role in optimizing cognitive processes, determining the selectivity and sustainability of individual behavior, and positively impacting the performance of learned behaviors and the emergence of new behaviors [26]. In a study on adolescents’ autonomous motivation to exercise, Palmer et al. found that a pro-autonomy environment and positive perceptions of physical activity are essential for increasing physical activity time [23]. Thus, it is evident that the satisfaction of basic psychological needs in exercise can serve as a source of incentive, influencing the interplay between cognition and behavior. However, it remains unclear whether the satisfaction of basic psychological needs for exercise, the focus of this study, can mediate the relationship between adolescents’ exercise cognition and exercise behaviors.

Previous research has significantly enhanced our understanding of the relationship between exercise cognition and behavior, verifying that individuals with higher exercise cognition engage more in exercise behavior. However, the majority of published research has focused on older age groups, particularly college students (e.g., the research of Kang et al. [14]). Given the important role of adolescents in future social development and the positive impact of exercise cognition and the satisfaction of basic psychological needs in exercise on their exercise behaviors, more attention should be directed towards studying the relationship between exercise cognition and the satisfaction of basic psychological needs in exercise among adolescents. Therefore, we posit the following hypotheses:

**Hypothesis 4**:
*Adolescents with high exercise cognition and satisfaction with basic psychological needs in exercise would be more likely to engage in exercise.*


**Hypothesis 5**:
*The adolescents’ satisfaction of basic psychological needs in exercise functions as a significant positive mediator between exercise cognition and exercise behavior.*


### 1.4. The Model of Study

To validate the above hypotheses, this study employed SEM with exercise cognition as the independent variable, exercise behavior as the dependent variable, and satisfaction of basic psychological needs in exercise as the mediating variable. The hypothesized model of the study is depicted in Figure 1.

## 2. Materials and Methods

### 2.1. Participants

A total of 1516 participants were recruited at the beginning of the study, and 996 participants provided usable data. Among them, 55.40% were females and 44.60% were males. The average age of the participants was 13.34 years (SD = 1.059).

### 2.2. Instruments

#### 2.2.1. Exercise Cognition

To measure adolescents’ exercise cognition, the College Students’ Exercise Cognition Scale [27] was revised to suit the current study’s purpose, considering the age and setting differences between middle school and college students. The research team revised the items based on empirical data collected from adolescents and insights from experts in the field. The final version of the scale was developed to effectively align with adolescents’ exercise cognition status. Specifically, the following five questions were modified: “You think exercise has a good impact on your academic study and life”, “You know how to participate in exercises”, “You are interested in exercises”, “You have a need for participating in exercises” and “You understand the hardship and pleasure of participation in exercises”. The internal consistency coefficient of the full scale was acceptable (i.e., α = 0.819) (see Table 1 and Table 2).

The composite reliability (CR) was 0.850, with evaluation criteria greater than 0.7, exceeding the significant threshold of 0.6 [28]. The AVE was greater than 0.5 (AVE = 0.531), and the evaluation criteria were greater than 0.5 [29], indicating that the questionnaire had good discriminant validity. The modified scale confirmatory factor analysis (CFA) fit indices were all within an acceptable range (i.e., χ^2^/df = 1.31, CFI = 0.99, NFI = 0.99, GFI = 0.99, AGFI = 0.99, RMSEA = 0.02), which were all within the acceptable range [30,31], indicating acceptable construct validity for the scale.

#### 2.2.2. Satisfaction of Basic Psychological Needs in Exercise

Wilson et al. developed a scale measuring psychological need satisfaction in exercise in 2006 [32]. This scale was later translated and revalidated by Wang and Yu in China [33] and has since been used to measure adolescents’ psychological need satisfaction in exercise. The scale has been utilized in numerous previous studies [34]. It consists of three domains: autonomy, competence, and relatedness, with four items in each domain. The internal consistency Cronbach’s α coefficients for the domains of autonomy, competence, relatedness, and the overall scale were 0.92, 0.91, 0.89, and 0.92, respectively, all exceeding the evaluation criterion of 0.7, indicating acceptable reliability [35]. The composite reliability (CR) for the domains of autonomy, competence, and relatedness was 0.850, 0.851, and 0.815, respectively, surpassing the significant threshold of 0.6 [28]. The average variance extracted (AVE) values were greater than 0.5 (autonomy = 0.590, competence = 0.589, relatedness = 0.526), suggesting acceptable discriminant validity for the questionnaire [29]. The confirmatory factor analysis (CFA) fit indices were all within the acceptable range (χ^2^/df = 2.44, CFI = 0.93, NFI = 0.96, GFI = 0.95, AGFI = 0.95, RMSEA = 0.07) [30,31], indicating acceptable construct validity for the scale (see Table 1 and Table 2).

#### 2.2.3. Exercise Behaviors

This study utilized the Adolescent Exercise Behaviors Scale developed by Hagger et al. [36], which was subsequently revised by Chinese scholars Wang L. and Zheng [37]. In line with the research objectives, three specific subscales of the scale were selected for this study: behavioral intention, perceived behavioral control, and habitual behavior. The scale included nine items measured using a 5-point Likert scale. The internal consistency Cronbach’s α coefficient for the domains of behavioral intention, perceived behavioral control, habitual behavior, and the overall scale were 0.714, 0.835, 0.792, and 0.792, respectively (see Table 2). These alpha values are all within the acceptable range. The composite reliability (CR) for the three domains was 0.890 (behavioral intention), 0.881 (perceived behavioral control), and 0.902 (habitual behavior), all exceeding the significant threshold of 0.6 [28]. The average variance extracted (AVE) was greater than 0.5 (behavioral intention = 0.731, perceived behavioral control = 0.712, and habitual behavior = 0.755), suggesting good discriminant validity [29]. The confirmatory factor analysis (CFA) fit indices for the modified scale were all within the acceptable range (i.e., χ^2^/df = 4.17, CFI = 0.91, NFI = 0.89, GFI = 0.98, AGFI = 0.96, RMSEA = 0.07) [30,31], indicating acceptable construct validity for the scale (see Table 1 and Table 2).

### 2.3. Data Collection and Analysis

Before collecting any data, the first author obtained institutional review board approval from his affiliated university. Based on the study’s objectives, the following previously developed or selected scales were used to collect the data. Some of the existing scales were revised to fit the needs of the study. The reliability and construct validity of the revised scales were re-examined, as suggested by Meyers et al. [38], to ensure acceptable reliability and validity within the context of the study’s sample. The construct validity of the survey was examined using confirmatory factor analysis (CFA) and structural equation modeling (SEM) conducted in IBM SPSS Amos 26.0 Graphics. Several statistical indices were evaluated to assess the model fit, including (a) the minimum discrepancy per degree of freedom (χ^2^/df), (b) the comparative fit index (CFI), (c) the normed fit index (NFI), (d) goodness of fit index (GFI), (e) adjusted goodness of fit index (AGFI), and (f) root mean square error of approximation (RMSEA). Following the guidelines outlined by Meyers et al., a good model fit is indicated by an RMSEA value of 0.08 or less, as well as a GFI, NFI, AGFI, and CFI value of 0.90 or higher [30,31]. Additionally, to establish a reasonable model fit, a χ^2^/df value of 5 or less is required [30,31].

The aforementioned scales were compiled into an online survey, which included common demographic questions at the end. The survey provided participants with the following information: (a) the purpose of the study, (b) instructions for completing the survey, and (c) confidentiality details.

The survey was then uploaded to the online platform “Wenjuanxing,” which is similar to Amazon Mechanical Turk. A cluster random sampling technique within classes was used to select participants. A link to the online survey was sent to all selected students by their teachers. Students independently and anonymously completed the questionnaire online using their own mobile phones or computers outside of class. Consequently, the instructors and parents of these participants remained unaware of the identities of those who took part in the study, ensuring participant confidentiality and the collection of honest responses.

Prior to conducting data analysis, a comprehensive data cleaning process was implemented. Cases containing more than 50% missing values were systematically excluded to uphold data integrity, and any extreme outliers were rigorously eliminated. The reliability of the scale’s scores was assessed by calculating Cronbach’s alpha using SPSS 24.0, with an acceptable alpha value defined as 0.70 or above, as suggested by Meyers et al. [39].

Fit testing was performed on the established SEM using Amos 24.0. The model fit indices—χ^2^/df = 2.413, RMSEA = 0.038, RMR = 0.019, GFI = 0.982, CFI = 0.985, and TLI = 0.980—indicate a good fit of the model. After 5000 resampling iterations, hypothesis testing was conducted by calculating 95% confidence intervals.

## 3. Results

### 3.1. Correlation Analysis of Adolescents’ Exercise Cognition, Satisfaction of Basic Psychological Needs in Exercise, and Exercise Behaviors

To test whether adolescents with higher exercise cognition and satisfaction of basic psychological needs in exercise are more likely to exercise behavior, a Pearson correlation analysis was conducted on the variables (see Table 3).

The results revealed that the average score for adolescents’ exercise cognition was 4.18 ± 0.621. Regarding the satisfaction of basic psychological needs in exercise, the scores for autonomy, competence, and relatedness were 3.77 ± 0.804, 3.169 ± 3.73, and 3.22 ± 0.790, respectively. The dimensions of exercise behavior, including behavioral intention, perceived behavioral control, and habit, scored 3.98 ± 0.830, 3.77 ± 0.861, and 3.45 ± 1.023, respectively. As shown in Table 4, all dimensions exhibited positive correlations, with correlation coefficients ranging from 0.117 to 0.753.

This indicates a moderate to high correlation between adolescents’ exercise cognition, satisfaction of basic psychological needs in exercise, and exercise behaviors, and moreover the correlation coefficient between exercise cognition, relationship satisfaction and exercise behaviors were higher than that between competence satisfaction, autonomy satisfaction, and exercise behaviors.

### 3.2. Gender Differences in All Variables 

To examine whether there are differences between boys and girls in terms of their exercise cognition, satisfaction of basic psychological needs in exercise, and their exercise behaviors, an independent samples *t*-test was conducted on each variable (see Table 4).

The study revealed statistically significant differences between different genders in the research variables, including exercise cognition (t = 3.35, df = 952.45, *p* < 0.01), competence (t = 3.89, df = 939.22, *p* < 0.01), and relatedness (t = 4.59, df = 952.76, *p* < 0.01) of the satisfaction of basic psychological needs in exercise, and exercise behaviors (t = 2.99, df = 921.92, *p* < 0.01). Specifically, male adolescents showed significantly higher levels of exercise cognition, sense of competence, sense of relatedness, and exercise behavior compared to their female counterparts.

### 3.3. Construction of SEM for Adolescents’ Exercise Cognition and Exercise Behaviors

To evaluate the direct, indirect, and overall effects of adolescents’ exercise cognition and basic psychological needs in exercise and exercise behaviors, a path analysis was conducted on the established SEM (see Table 5 and Figure 2).

As shown in Table 5 and Figure 2, the impact coefficient of exercise cognition on exercise behaviors was 0.276 (*p* < 0.001, 95% CI [0.125, 0.426], excluding 0). The path coefficient of satisfaction of exercise basic psychological needs in exercise behaviors was 0.617 (*p* < 0.001, 95% CI [0.447, 0.812], excluding 0). The mediating effect of satisfaction of basic psychological needs in exercise in the relationship between exercise cognition and exercise behaviors was 0.401 (*p* < 0.001, 95% CI [0.288, 0.544], excluding 0).

In summary, Hypotheses were confirmed, indicating that satisfaction of basic psychological needs in exercise mediates the impact of exercise cognition on exercise behaviors, accounting for 59.24% of the total effect. This implies that 59.24% of the total effect of exercise cognition on exercise behaviors was achieved through the mediation of satisfaction of basic psychological needs in exercise, indicating a partial mediating role in the influence of exercise cognition on exercise behavior.

## 4. Discussion

### 4.1. The Relationship between Adolescents’ Exercise Cognition and Exercise Behaviors

The results indicated a positive correlation between adolescents’ exercise cognition and exercise behaviors. Additionally, it was found that adolescents’ exercise cognition positively influences their exercise behaviors. Thus, hypotheses 1 and 2 were confirmed. The analysis of the data from the current study confirms that exercise cognition enhances adolescents’ participation in physical activity. The higher the level of adolescents’ exercise cognition, the greater the likelihood of their engaging in exercise behaviors. In other words, a higher level of exercise cognition leads to more active participation in physical activity among adolescents in a complex daily life environment. Conversely, incorrect and negative exercise cognition can hinder adolescents’ exercise behaviors. Therefore, in daily life, adolescents with positive exercise cognition (e.g., believing that exercise is beneficial) are more likely to engage in exercise behaviors and develop good exercise habits. This finding aligns with the results reported by Adams, who found that the exercise system adheres to the instructions of the cognitive system [40]. Individuals, upon reflecting on their behavior, are able to systematically utilize all available information to determine whether to adopt a behavior [41].

Despite the well-documented importance of physical activity as a health-promoting behavior, many people remain unaware of its benefits for physical health, mental health, cognition, and learning [42,43,44]. Globally, less than 20% of adolescents aged 11–17 years meet the recommended standard of 60 min of physical activity per day [45]. Therefore, promoting exercise behavior among adolescents is a public health priority both in China and worldwide [1,46]. To effectively encourage exercise behavior among adolescents, it is crucial to understand its determinants. Enhancing adolescents’ exercise cognition is essential for promoting their future exercise behavior.

### 4.2. Gender Differences in Exercise Cognition, Satisfaction of Basic Psychological Needs in Exercise, and Their Exercise Behaviors

The results of the study support the hypothesis that there are sex differences in exercise cognition, the satisfaction of basic psychological needs for exercise, and exercise behavior. Specifically, there were notable gender differences between male and female adolescents in terms of exercise cognition, basic psychological needs for exercise, and exercise behavior, consistent with previous studies [43,44]. Although physical activity promotes physical health among both male and female adolescents, the value and purpose of physical activity differ between the sexes. Boys tended to focus on the health, competence, and social motivations of physical activity, while girls were more concerned with its effects on weight and appearance [45,46].

When boys receive praise for improving their speed and strength through physical activities or making more friends, they are more likely to experience psychological satisfaction from physical activity. In contrast, the satisfaction of girls’ basic psychological needs for physical activity comes primarily from achieving weight loss and improving body shape, consistent with the findings of Beville et al. [47,48]. Therefore, to effectively motivate male and female adolescents to participate in physical activities, it is critical to offer a variety of activities that cater to the different exercise behavior needs of each gender.

### 4.3. The Mediating Role of Satisfaction of Basic Psychological Needs in Exercise between Exercise Cognition and Exercise Behaviors among Adolescents

The results of the study demonstrated a positive relationship between the satisfaction level of adolescents’ basic psychological needs for exercise and their exercise behavior. It was found that the satisfaction level of these needs positively influences their exercise behavior, supporting hypotheses 3 and 4. Hypothesis 5 was also confirmed, as the satisfaction level of adolescents’ basic psychological needs for exercise significantly mediates the relationship between exercise cognition and exercise behavior. When adolescents’ basic psychological needs for exercise are satisfied, there is a greater likelihood of their participation in exercise behavior. The degree of satisfaction of these needs mediates the relationship between exercise cognition and exercise behavior. Higher levels of exercise cognition lead to higher satisfaction with basic psychological needs for exercise, which in turn increases the likelihood of participation in physical activity. The mediating effect may be attributed to the fact that higher exercise cognition fosters a stronger sense of exercise autonomy, relatedness, and competence. This satisfaction of basic psychological needs enhances the likelihood of adolescents participating in exercise and positively contributes to the emergence and development of individual behaviors when exercise-related autonomy, relatedness, and competence are satisfied [22,49].

French and Lin [50,51] found through their research that adding emotional support between cognition and behavior significantly enhances the occurrence of behavior because emotional factors can provide motivational stimuli that influence behavior. Therefore, the satisfaction of psychological needs, as an emotional factor, plays a crucial role in the transition from cognition to behavior [52]. Good exercise cognition is a prerequisite for the satisfaction of basic psychological needs for exercise, and fulfilling these needs can lead to stronger behavioral intentions to participate in exercise and the development of good exercise habits [53]. Therefore, it is essential to conduct detailed investigations and understand adolescents’ sense of autonomy, relatedness, and competence in the context of physical activities to meet their three basic psychological needs for exercise. By satisfying these psychological needs, adolescents’ exercise behaviors can be enhanced, leading to the development of good exercise habits.

## 5. Limitations

The sample of the current study was mainly from Henan and Hebei province in China; therefore, cautions need to be exercised when generating the results to other populations. Moreover, the measurement of variables in our study was self-reported, which may have some unavoidable biases. Culture effect on the exercise cognition was not the focus of the current study and it should be studied in the future.

## 6. Conclusions

The findings suggest that positive exercise cognition enhances students’ behavioral intentions and habits to engage in physical activity. Additionally, the satisfaction of basic psychological needs for exercise mediates the relationship between exercise cognition and adolescent exercise behaviors. This study contributes to the existing literature by offering novel insights and guidance for schools and society on strategies to enhance adolescents’ exercise behaviors.

## Figures and Tables

**Figure 1 behavsci-14-00574-f001:**
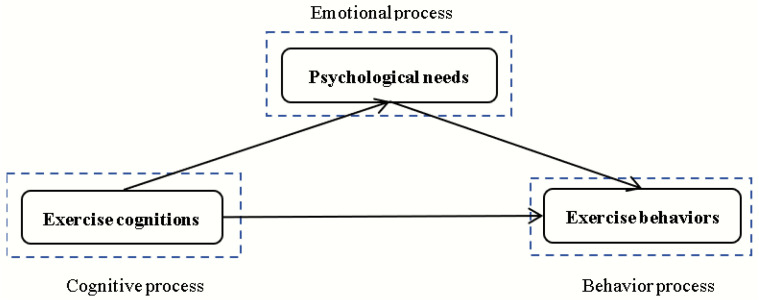
Hypothetical model.

**Figure 2 behavsci-14-00574-f002:**
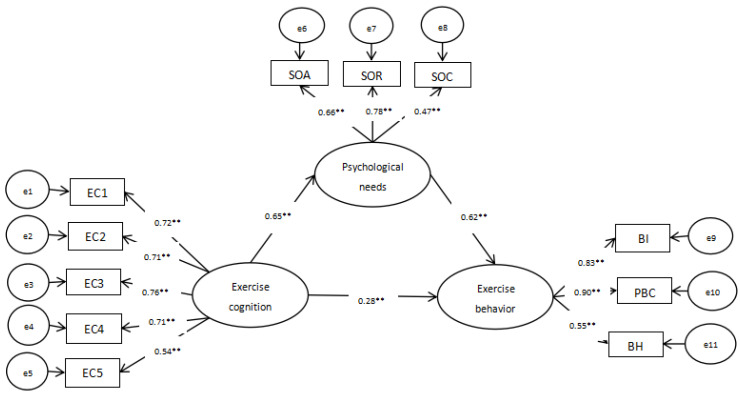
Path analysis of SEM. ** *p* < 0.001.

**Table 1 behavsci-14-00574-t001:** Fitting indexes of confirmatory factor analysis.

Factor	χ^2^/df	CFI	NFI	GFI	AGFI	RMESA
Exercise cognition	1.31	0.99	0.99	0.99	0.99	0.02
Satisfaction of basic psychological needs in exercise	2.44	0.93	0.96	0.95	0.95	0.07
Exercise behavior	4.17	0.91	0.89	0.98	0.96	0.07
Exercise cognition←→Satisfaction of basic psychological needs in exercise←→Exercise behavior	2.413	0.98	0.97	0.98	0.97	0.03

**Table 2 behavsci-14-00574-t002:** Reliability and convergent validity of the study constructs.

Construct	Item	Standard Loading	CR	Cronbach’s α	AVE
Exercise cognition (EC)	EC1	0.750	0.850	0.819	0.531
	EC2	0.706			
	EC3	0.740			
	EC4	0.727			
	EC5	0.722			
Sense of autonomy (SOA)	SOA1	0.619	0.850	0.920	0.590
	SOA2	0.803			
	SOA3	0.848			
	SOA3	0.784			
Sense of competence (SOC)	SOC1	0.658	0.815	0.890	0.526
	SOC2	0.767			
	SOC3	0.769			
	SOC4	0.702			
Sense of relatedness (SOR)	SOR1	0.804	0.851	0.910	0.589
	SOR2	0.754			
	SOR3	0.725			
	SOR4	0.785			
Behavioral intention (BI)	BI1	0.852	0.890	0.714	0.731
	BI2	0.832			
	BI3	0.881			
Perceived behavioral control (PBC)	PBC1	0.872	0.881	0.835	0.712
	PBC2	0.881			
	PBC3	0.776			
Behavioral habits (BH)	BH1	0.890	0.902	0.792	0.755
	BH2	0.873			
	BH3	0.843			

**Table 3 behavsci-14-00574-t003:** Correlation analysis results.

Variables	M ± SD	EC	SOA	SOC	SOR	BI	PBC	BH
EC	4.18 ± 0.621	1						
SOA	3.77 ± 0.804	0.417 **	1					
SOC	3.169 ± 3.73	0.262 **	0.335 **	1				
SOR	3.22 ± 0.790	0.500 **	0.509 **	0.377 *	1			
BI	3.98 ± 0.830	0.441 **	0.358 **	0.240 **	0.411 **	1		
PBC	3.77 ± 0.861	0.440 **	0.397 **	0.263 **	0.439 **	0.753 **	1	
BH	3.45 ± 1.023	0.320 **	0.239 **	0.117 **	0.311 **	0.437 **	0.501 **	1

Note: * *p* < 0.05; ** *p* < 0.01; EC—exercise cognition; SOA—sense of autonomy; SOR—sense of competence; SOC—sense of relatedness; BI—behavioral intention; PBC—perceived behavioral control; BH—behavioral habits.

**Table 4 behavsci-14-00574-t004:** Analysis of gender differences in research variables.

	Category	Research Variables				
		Exercise Cognition	Sense of Autonomy	Sense of Competence	Sense of Relatedness	Exercise Behaviors
Gender	Male	4.17 ± 0.66	3.80 ± 0.78	3.29 ± 0.88	3.35 ± 0.77	3.81 ± 0.78
	Female	4.02 ± 0.67	3.75 ± 0.81	3.07 ± 0.86	3.12 ± 0.78	3.67 ± 0.73
*t*		3.35	0.939	3.89	4.59	2.99
*p*		0.001 **	0.348	0.000 **	0.000 **	0.003 **

Note: ** *p* < 0.01.

**Table 5 behavsci-14-00574-t005:** Hypotheses model path and effect decomposition.

Direct Effect	Path	B	S.E.	C.R.	95% Confidence Interval	Effect Ratio
Direct effect	Exercise cognition → Exercise behaviors	0.276 **	0.067	4.142	0.125, 0.426	40.76%
	Exercise cognition → Satisfaction of basic psychological needs in exercise	0.650 **	0.046	14.225	0.560, 0.750	
	Satisfaction of basic psychological needs in exercise → Exercise behaviors	0.617 **	0.079	7.761	0.447, 0.812	
Intermediary effect	Exercise cognition → Satisfaction of basic psychological needs in exercise → Exercise behaviors	0.401 **			0.288, 0.544	59.24%

Note: ** *p* < 0.001.

## Data Availability

The anonymized data that support the findings of this study are available on request from the corresponding author. The data are not publicly available due to containing information that may comprise the participants’ privacy.

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
