# Peer review of "The Impact of Exercise Cognition on Exercise Behaviors: The Mediating Role of the Satisfaction of Basic Psychological Needs in Exercise for Adolescents"

_behavsci, 2024, doi:10.3390/bs14070574_

Round 1

Reviewer 1 Report

Comments and Suggestions for Authors

The author(s) mentioned that cognition can be divided into two categories, providing descriptions for each. However, based on the literature cited in this paper (e.g., references 10, 11, 12), it was challenging to find clear evidence supporting this categorization. Clarifying this aspect of the study would be beneficial, because I believe this is an important aspect of the study. 

The previous studies referenced appear to have conducted studies based on definitions different from 'sport cognition' as defined in this paper. In other words, to support the author(s)' argument and explanation about sport cognition, it would be beneficial to reference studies that have an identical construct or similar construct or definition of sport cognition as defined in this study. This issue persists not only in defining sport cognition but also in describing the relationships between different constructs. Thus, it appears that the literature review in this manuscript is insufficient. 

The literature review suddenly introduces 'gender' without sufficient rationale. Providing a rationale for why gender differences should be considered would be beneficial.

In section 1.3, the authors attempt to discuss mediation effects. However, some studies do not align with the authors' descriptions (e.g., reference 25).

Hypotheses

Typically, hypotheses are often mentioned in the literature review section because they are developed based on the literature review. So, I suggest the author(s) reconsider this once more. 

The primary purpose of this study adds further confusion. The literature review does not provide sufficient evidence or rationale for Hypotheses 1 and 2. For example, the authors mention that the primary goal is to examine gender differences. If this is the case, the literature review should adequately address this issue.

The order of the sub-sections (or explanation) in the Methods could be adjusted. Usually, descriptions about data collection and measurement methods are provided in the method section, and the reliability and validity of measurement tools are typically discussed in the results section. Following this approach could help readers understand the manuscript better.

There are many issues and omissions in the method section. Generally, conducting a Confirmatory Factor Analysis (CFA) including all constructs together is a better approach to verifying validity than running it based on individual constructs.

Furthermore, it is recommended that the authors provide references regarding the thresholds mentioned for data analysis. 

There is a need for further work and reporting on validity. While the authors have discussed convergent validity, they have not addressed discriminant validity. Including this could significantly improve the quality of the manuscript.

Reviewing the Discussion/Limitation section based on the resolution of the aforementioned issues would be a good approach. The current discussion appears weak, therefore, a more in-depth discussion related to previous research and practical implications should be more elaborately described.

Comments on the Quality of English Language

Even though no serious issues were found, it is apparent from reading this manuscript that it was written by someone not fluent in English. 

Author Response

Thank you to the three reviewers for their valuable suggestions on improving the manuscript. We have responded to each comment and made most of the suggested changes. We also explained why some changes were not made. The paper was also proofread to avoid grammatical errors.

Review#1

Comment 1. The author(s) mentioned that cognition can be divided into two categories, providing descriptions for each. However, based on the literature cited in this paper (e.g., references 10, 11, 12), it was challenging to find clear evidence supporting this categorization. Clarifying this aspect of the study would be beneficial, because I believe this is an important aspect of the study.

Response 1: We really appreciate your comments on this matter. However, we have revised the entire paragraph to focus on exercise cognition give that sport cognition has not been examined thoroughly in the literature and is not the focus of the current study. We believe that exercise cognition focuses on physical activities that are for the purpose of health and fitness, which is consistent with the content of this study. As such, we believe it is appropriate to use of the concept of exercise cognition throughout this paper.

The references have also been replaced with the ones related to exercise cognition. Please see L175-L178.

Comment 2. The previous studies referenced appear to have conducted studies based on definitions different from 'sport cognition' as defined in this paper. In other words, to support the author(s)' argument and explanation about sport cognition, it would be beneficial to reference studies that have an identical construct or similar construct or definition of sport cognition as defined in this study. This issue persists not only in defining sport cognition but also in describing the relationships between different constructs. Thus, it appears that the literature review in this manuscript is insufficient.

Response 2: New references have been used in the literature review section, and the literature review for that section has been revised, please see L178-L189.

Comment 3. The literature review suddenly introduces 'gender' without sufficient rationale. Providing a rationale for why gender differences should be considered would be beneficial.

Response 3: We have added an introduction to the issue of gender. Please see L215-L224.

Comment 4. In section 1.3, the authors attempt to discuss mediation effects. However, some studies do not

align with the authors' descriptions (e.g., reference 25).

Response 4: We revised the references, please see L232-L235.

Comment 5. Typically, hypotheses are often mentioned in the literature review section because they are developed based on the literature review. So, I suggest the author(s) reconsider this once more.

Response 5: Done, please see L190 (Hypothesis 1), L225-L228 (Hypothesis 2 and Hypothesis 3), L256-L259(Hypothesis 4 and Hypothesis 5).

Comment 6. The primary purpose of this study adds further confusion. The literature review does not provide sufficient evidence or rationale for Hypotheses 1 and 2. For example, the authors mention that the primary goal is to examine gender differences. If this is the case, the literature review should adequately address this issue.

Response 6: Wed have revised literature review section to provide the rationale for hypotheses 1 and 2. Please see L215-L224.

Comment 7. The order of the sub-sections (or explanation) in the Methods could be adjusted. Usually, descriptions about data collection and measurement methods are provided in the method section, and the reliability and validity of measurement tools are typically discussed in the results section. Following this approach could help readers understand the manuscript better.

Response 7: We have thought about it carefully and respectfully disagree with this comment. In this journal, the reliability and validity of measurement tools are mostly presented in the section introducing measurement tools. For examples:

[1] Sease, T.B.; Sandoz, E.K.; Yoke, L.; Swets, J.A.; Cox, C.R. Loneliness and Relationship Well-Being: Investigating the Mediating Roles of Relationship Awareness and Distraction among Romantic Partners. Behav. Sci. 2024, 14, 439. https://doi.org/10.3390/bs14060439

[2] Olave, L.; Iruarrizaga, I.; Herrero, M.; Macía, P.; Momeñe, J.; Macía, L.; Muñiz, J.A.; Estevez, A. Exercise Addiction and Intimate Partner Violence: The Role of Impulsivity, Self-Esteem, and Emotional Dependence. Behav. Sci. 2024, 14, 420. https://doi.org/10.3390/bs14050420

[3] Li, M.; Liu, F.; Yang, C. Teachers’ Emotional Intelligence and Organizational Commitment: A Moderated Mediation Model of Teachers’ Psychological Well-Being and Principal Transformational Leadership. Behav. Sci. 2024, 14, 345. https://doi.org/10.3390/bs14040345

Comment 8. There are many issues and omissions in the method section. Generally, conducting a Confirmatory Factor Analysis (CFA) including all constructs together is a better approach to verifying validity than running it based on individual constructs.

Response 8: We have added it to Table 1. Please see L284.

Comment 9. Furthermore, it is recommended that the authors provide references regarding the thresholds mentioned for data analysis.

Response 9: The data analysis thresholds was added, e.g. composite reliability (CR) was 0.850, evaluation criteria greater than 0.7. References were added. Please see L286-L291.

Comment 10. There is a need for further work and reporting on validity. While the authors have discussed convergent validity, they have not addressed discriminant validity. Including this could significantly improve the quality of the manuscript.

Response 10: This study reported discriminant validity (AVE) in Table2. Please see L285.

Comment 11. Reviewing the Discussion/Limitation section based on the resolution of the aforementioned issues would be a good approach. The current discussion appears weak, therefore, a more in-depth discussion related to previous research and practical implications should be more elaborately described.

Response 11: The discussion section was revised accordingly and a separate discussion of gender differences was added. Please see L421-L499.

Comment 12. Even though no serious issues were found, it is apparent from reading this manuscript that it was written by someone not fluent in English.

Response 12: We have asked an American professor to proofread it.

Reviewer 2 Report

Comments and Suggestions for Authors

The main purpose of this research is to determine gender differences in adolescents’ sports cognition, satisfaction of basic psychological needs in exercise, and exercise behavior. The secondary purpose of this study was to use a structural path analysis model to assess the direct, indirect, and total effects of adolescents’ sports cognition and basic psychological needs in exercise, and exercise behaviors. 

The current study’s hypotheses based on 996 valid questionnaires collected from adolescents in China. The Results indicated that positive direct impact of adolescents’ sports cognition was found on exercise behavior. In addition, the satisfaction of basic psychological needs in exercise played a partial  mediating role in the relationship between sports cognition and exercise behavior.

I believe that this is an interesting study and there is a merit for this journal. This study designed well and it has validity and reliability as well with bigger sample size. However, there are minor issues. First, a better abstract needed with clear number and results. Second, discussion and conclusion parts are weak and need improvement with detailed comparison with current research. Lastly, a practical suggestions for the future need to be added. Also I wonder how culture have an effect on the sport cognition because study completed in an international country.

I would like to thank authors for this interesting study and I look forward to seeing edited version of this manuscript. Best regards,

Author Response

Thank you to the three reviewers for their valuable suggestions on improving the manuscript. We have responded to each comment and made most of the suggested changes. We also explained why some changes were not made. The paper was also proofread to avoid grammatical errors.

Review#2

Comment 1. I believe that this is an interesting study and there is a merit for this journal. This study designed well and it has validity and reliability as well with bigger sample size.

Response 1: Thank you!

Comment 2. However, there are minor issues. First, a better abstract needed with clear number and results.

Response 2: The abstract has been revised to clarify the figures and results. Please see L133-L148.

Comment 3. Second, discussion and conclusion parts are weak and need improvement with detailed comparison with current research.

Response 3: The discussion and conclusion have been revised. Please see L421-L4933 and L500-L506. In the literature review section, "exercise cognition" was rewritten to define and introduce. Please see L175-L178.

Comment 4. Lastly, a practical suggestion for the future need to be added. Also, I wonder how culture have an effect on the sport cognition because study completed in an international country.

Response 4: Practical suggestions for the future have been added. Culture effect on the sport cognition was not the focus of the current study and it should be studied in the future. Please see L494-L499.

Reviewer 3 Report

Comments and Suggestions for Authors

Overview

The authors examined the relationship between sports cognition, satisfaction of basic psychological needs in exercise, and exercise behavior among Chinese adolescents. 

The manuscript is well written. I have some comments to make to improve it.

Specific comments

Abstract

Line 18: Replace with "Verified".

Keywords

Remove keywords from the title. Replace keywords with ones other than the title. This will allow you to optimize the search for your manuscript after publication.

Introduction

Put paragraph 2.1 "Research questions and hypotheses" in the Introduction section.

Materials and methods

Begin this section with "The Hypothesized Model of Study".

The section is clearly written and the statistics used are appropriate.

Results

This section correctly describes the results from the statistical analyzes carried out.

Discussion

Please rewrite the discussion section describing whether each of the hypotheses formulated were confirmed or refuted by results.

After the “Limitations” section, insert a small conclusion paragraph to briefly restate your main findings (the take-home message) and the novelty of the paper according to the current literature to help better readers understanding how this paper is different from other already published

Comments on the Quality of English Language

No comment.

Author Response

Thank you to the three reviewers for their valuable suggestions on improving the manuscript. We have responded to each comment and made most of the suggested changes. We also explained why some changes were not made. The paper was also proofread to avoid grammatical errors.

Review#3

Comment 1. Line 18: Replace with "Verified".

Response 1: Done, please see L138.

Comment 2. Remove keywords from the title. Replace keywords with ones other than the title. This will allow you to optimize the search for your manuscript after publication.

Response 2: Done, please see L149.

Comment 3. Put paragraph 2.1 "Research questions and hypotheses" in the Introduction section.

Adjusted the location of the research hypotheses by placing them below the literature review for each variable.

Response 3: Done, please see L190 (Hypothesis 1), L225-L228 (Hypothesis 2 and Hypothesis 3), L256-L259(Hypothesis 4 and Hypothesis 5).

Comment 4. Please rewrite the discussion section describing whether each of the hypotheses formulated were confirmed or refuted by results.

Response 4: Done. Please see L422-L494.

Comment 5. After the “Limitations” section, insert a small conclusion paragraph to briefly restate your main findings (the take-home message) and the novelty of the paper according to the current literature to help better readers understanding how this paper is different from other already published.

Response 5: Done. Please see L501-L507.
